# A Joint Acoustic Emission Source Localization Method for Composite Materials

**DOI:** 10.3390/s23125473

**Published:** 2023-06-09

**Authors:** Xiaoran Wang, Fang Yin, Zhishuai Wan

**Affiliations:** 1Faculty of Information Technology, Beijing University of Technology, Beijing 100124, China; 2Beijing Institute of Control Engineering, Beijing 100076, China; 3Institute of Advanced Structure Technology, Beijing Institute of Technology, Beijing 100081, China

**Keywords:** composite materials, acoustic emission, time-difference-blind localization, beamforming localization, joint localization

## Abstract

Damage localization methods for composite materials are a popular research topic at present. The time-difference-blind localization method and beamforming localization method are often individually utilized in the localization of the acoustic emission sources of composite materials. Based on the performances of the two methods, a joint localization method for the acoustic emission sources of composite materials is proposed in this paper. Firstly, the performance of the time-difference-blind localization method and the beamforming localization method were analyzed. Then, with the advantages and disadvantages of these two methods in mind, a joint localization method was proposed. Finally, the performance of the joint localization method was verified using simulations and experiments. The results show that the joint localization method can reduce the localization time by half compared with the beamforming localization method. At the same time, compared with the time-difference-blind localization method, the localization accuracy can be improved.

## 1. Introduction

Due to their advantages of high strength, high stiffness and material performance designability, composite materials are increasingly used in wind power generation, aerospace civil engineering, automobile industry, ocean engineering, pressure vessels and other fields [1,2,3,4,5,6]. For example, in the wind power industry, glass-fiber-reinforced composites have become the material of choice for most large wind power blades [3]. As far as aircraft are concerned, composite materials are increasingly being used in aircraft structures due to their lightweight and robust properties [7]. The amount of carbon fiber composite materials used in the European Airbus A380 and A350-XWB large civil airliners reached 25% and 53% of the total weight of the structure, respectively. However, even with careful design and sophisticated manufacturing processes, composite materials can still be damaged during manufacture, use and maintenance. Since most of these damages occur in the interior of the composite structures, whose internal composition is quite complex and difficult to locate from its appearance, the damage detection of composite structures is extraordinarily difficult. Currently, many nondestructive testing methods have been proposed for the damage detection of composite materials, such as penetration methods and ultrasonic methods, but these techniques have their own advantages, disadvantages and limitations. However, acoustic emission (AE) technology exhibits obvious advantages over other nondestructive testing technologies in terms of damage localization, which is one of the key points of damage detection. AE technology can provide dynamic information regarding the damage under the stress from the structure being loaded and assess the occurrence, expansion and location of damage within the structure online in real time. Meanwhile, AE technology possesses high sensitivity and accuracy. If the positions of the instruments and sensors are well arranged, then minor damage will be detected. Therefore, AE detection technology has important application prospects in the field of detection and localization of damage in composite materials [8].

Damage localization is an important step in the management of structural health; therefore, AE source localization technology has been extensively studied. The research on AE source localization technology can mainly be divided into the following categories: in the first category, AE localization is carried out by analyzing the modal acoustic emission (MAE), which typically requires the identification of the arrival times of extensional and flexural wave modes. For example, Surgeon and Wevers [9] proposed a linear localization method using two wave modes and one sensor, and they conducted feasibility verification on crossed and unidirectional carbon-fiber-reinforced polymer multilayer plates. MAE has also been implemented with more than one sensor. Qiu et al. [10] presented a damage source localization method suitable for understanding the fracture behavior of asphalt mixtures using AE detection. However, this kind of AE source localization technology through the study and analysis of waves is suitable for simple problems and also requires necessary mathematical skills, so it is not widely used.

The second category is the beamforming technique. Beamforming is a signal processing technique used in sensor arrays for directional signal transmission or reception [11]. McLaskey et al. [12] introduced the beamforming technique for AE source localization within civil structures. He et al. [13] improved the technique and extended it to AE source localization of plate structures. Nakatani et al. [14] utilized the beamforming technique for the localization of AE sources in anisotropic structures. He et al. [15] revealed the influence of AE propagation characteristics on the localization accuracy of the beamforming technique and combined plate wave theory and wavelet transform to propose a method for determining the wave velocity used in localization. He et al. [16] introduced the Hilbert curve to minimize the cost and maximize the computational performance of AE beamforming. The efficiency could be improved approximately 154 times as compared to the traditional beamforming method. Wang et al. [17] proposed a joint localization method based on beamforming and time difference of arrival. Both the simulation and experimental results demonstrated the improved accuracy of the proposed method and reduced the amount of calculation. Although the beamforming technique exhibits high localization accuracy and robustness of the algorithm, its inherent disadvantage is low computational efficiency, and it is affected by the array size and sensor spacing [18].

The third category is the time-difference-of-arrival (TDOA) method. Tobias [19] is one of the pioneers in the field of AE source localization for structural damage. He developed the TDOA method for structural sound source localization. The accurate localization of the AE source using TDOA requires two conditions: the accurate arrival time and the accurate wave velocity. However, AE is often affected by noise, energy attenuation, dispersion and other factors during propagation, and it is sometimes difficult to obtain accurate wave velocity and time difference [20,21,22]. In terms of improving the accuracy of AE source localization, Gollob et al. [23] suggested a method based on the anisotropic velocity model to locate the AE source in the structure of heterogeneous materials. This method is more accurate and reliable. Aiming at the problem that the accuracy of the initial time affects the accuracy of source localization, Madarshahian et al. [24] proposed a method to automatically select the most probable initial time using two competing methods within the framework of Bayesian theory. Dong et al. [25] proposed a collaborative localization method for finding the best source location by using analytical and iterative methods. The properties of the material also have a great influence on the localization of the AE source. Methods of locating the AE source without measuring or knowing the material properties is also a research hotspot. Kundu et al. [26] proposed a localization technique that can locate the sound source in a large anisotropic plate when the relationship between the velocity and the propagation direction is unknown. This technology only uses six sensors, which are arranged as two L-shaped sensor clusters. Yin et al. [27] improved the localization method proposed by Kundu by replacing two L-shaped sensor clusters with two Z-shaped sensor clusters and obtained more reliable results despite the addition of two sensors. Park et al. [28] introduced the assumption of aspheric waves to solve the problem of anisotropic plates exhibiting different dispersion curves in different propagation directions. Sen et al. [29,30] proposed a new square-shaped sensor cluster arrangement for AE localization in anisotropic plates.

Among these AE localization methods, the beamforming method based on the delay-and-sum algorithm is accurate in localization. However, it is necessary to divide the plate into meshes according to the required accuracy and then perform scanning calculations. This results in a higher calculation cost and a lower localization speed. The scheme that uses six sensors in an anisotropic plate (the time-difference-blind localization method proposed by Kundu [26]) presents many advantages, such as not needing to know the wave velocity and material properties, and the localization speed is exceptionally fast. Based on the performances of the time-difference-blind localization method and beamforming localization method, this paper proposes a joint localization method. The proposed method not only improves the accuracy of the time-difference-blind localization method, but also greatly reduces the localization calculation cost of the beamforming localization method.

The following chapters are arranged as follows: Section 2 introduces the time-difference-blind localization method and analyzes its performance. In Section 3, the localization performance of the beamforming localization method is analyzed. Section 4 proposes the joint localization method, and the method is verified by simulation signals. Section 5 carries out experimental verification. Finally, the summary and conclusion are given in Section 6.

## 2. Principle and Performance Analysis of Time-Difference-Blind Localization Method

### 2.1. Principle of Time-Difference-Blind Localization Method

The time-difference-blind localization method in the literature [26] works by arranging three sensors in an isosceles right-angle triangle, as shown in Figure 1. *d* is the length of a side of an isosceles right-angle triangle. If the coordinates of the three sensors S_1_, S_2_ and S_3_ are (*x*_1_, *y*_1_), (*x*_2_, *y*_2_) and (*x*_3_, *y*_3_), it is clear that
(1)$$\{\begin{array}{c} x_{2}=x_{1}+d\\ x_{3}=x_{1}\\ y_{2}=y_{1}\\ y_{3}=y_{1}+d \end{array}$$

The coordinate of the acoustic source A is (*x*_A_, *y*_A_). The distance between the sensors is much smaller than the distance *D* between the acoustic source A and the sensor S_i_. Therefore, the inclination angle *θ* of lines AS_1_, AS_2_ and AS_3_ should be roughly the same. Because of this assumption, the received signals at these three sensors will be almost identical but slightly time-shifted. The wave velocity in the direction from source point A to sensors S_1_, S_2_ and S_3_ should be almost similar, even for an anisotropic plate. Angle *θ* can be expressed as
(2)$$\theta =\tan^{-1}\left(\frac{y_{1}-y_{A}}{x_{1}-x_{A}}\right)\approx \tan^{-1}\left(\frac{y_{2}-y_{A}}{x_{2}-x_{A}}\right)\approx \tan^{-1}\left(\frac{y_{3}-y_{A}}{x_{3}-x_{A}}\right)$$

After the wave arrives at sensor S_1_, the time required for the waves to reach sensors S_2_ and S_3_ can be represented as *t*_21_ and *t*_31_, respectively. If *t*_21_ = *t*_2_ − *t*_1_ and *t*_31_ = *t*_3_ − *t*_1_, then
(3)$$\;\;t_{21}=\frac{d\cos\theta }{c\left(\theta \right)}$$
(4)$$t_{31}=\frac{d\sin\theta }{c\left(\theta \right)}$$
where c(*θ*) is the wave velocity in the direction of *θ.* From Equations (4) and (5), one can easily obtain
(5)$$\theta =\tan^{-1}\left(\frac{t_{31}}{t_{21}}\right)$$

It can be found that the AE source must be on the line through sensor S_1_ and which makes an angle *θ* with a horizontal line. Therefore, as long as two sets of sensors are arranged, two straight lines can be obtained, and the intersection of the two straight lines is the position of the AE source.

### 2.2. Test Object and Localization Results

The applicable conditions and performance of the time-difference-blind localization method based on an ideal anisotropic plate are studied. It is assumed that the velocity of waves propagating in the plate is a function of the angle and the velocity difference along the *x*-axis and *y*-axis is the largest [14]. The wave velocities propagation along the *x*-axis and *y*-axis are *Vx* and *Vy*, respectively. Then, the wave velocity in the direction of the α-angle to the *x*-axis direction can be expressed as Equation (6). The acoustic emission wave not only has velocity in both x and y directions, but the velocity at any point is also a function of the angle.
(6)$$V_{\alpha }=\sqrt{{\left(V_{x}\cos\alpha \right)}^{2}+{\left(V_{y}\sin\alpha \right)}^{2}}$$

In order to investigate the performance of the time-difference-blind localization method, eight different sets of *Vx*:*Vy* ratios were set in this paper to represent the strength of anisotropy. They were 1, 1.1, 1.2, 1.4, 1.5, 1.6, 1/1.4 and 1/1.6, and *Vy* = 5000 m/s was assumed. The size of the plate was 500 mm × 500 mm. Six sensors were set on the plate and the positions were (100 mm, 50 mm), (50 mm, 100 mm), (100 mm, 100 mm), (400 mm, 50 mm), (400 mm, 100 mm) and (450 mm, 100 mm). The specific values of the simulated AE source coordinates are shown in the first column of Table 1, where there are nine simulated AE sources. The approximate positions of the AE sources and sensors are shown in Figure 2 (triangles are the location of the acoustic sources, and the circles are the location of the sensors). Using the time-difference-blind localization method, the error is shown in Table 1. Figure 3 shows the localization results of different AE sources under different *Vx*:*Vy* values. In Figure 3, the results of *Vx*:*Vy* = 1.5 and *Vx*:*Vy* = 1.6 are missing from the localization result of the AE source at (250 mm, 450 mm), because the localization errors of *Vx*:*Vy* = 1.5 and *Vx*:*Vy* = 1.6 are too large to be shown in the figure. Then, *Vx*:*Vy* = 1 is assumed to be constant, and the sensor spacing was changed. The localization results are shown in Table 2.

The following can be seen from the results of different *Vx*:*Vy* values:

When the *Vx*:*Vy* value is greater than 1, the localization error increases with the increase of the *Vx*:*Vy* ratio. When the *Vx*:*Vy* value is smaller than 1, the localization error increases with the decrease of the *Vx*:*Vy* ratio. This means that the localization error increases when the *Vx*:*Vy* value is far from 1.

When the *Vx*:*Vy* value is near 1, the localization of the middle part of the plate is more accurate than the localization of both sides of the plate. Conversely, when the *Vx*:*Vy* value is far from 1, the localization of both sides of the plate is more accurate than the middle part of the plate.

Localization error increases with the increase of the distance between the acoustic source and the sensor.

The x-coordinates located by this localization method are accurate regardless of the *Vx*:*Vy* value. From Table 2, it can be found that the localization error of the time difference localization method increases with the increase of the sensors spacing.

When the *Vx*:*Vy* value is 1, the plate can be regarded as an isotropic plate. It can be seen that there are some errors from the results of the localization, which shows that there is an error in this method. The reasons for the error are analyzed below.

### 2.3. Error Analysis

From the principle of the time-difference-blind localization method, it can be determined that the method must meet the following geometrical relations.
(7)$$\{\begin{array}{c} d\cos\theta =D_{2}-D\\ d\sin\theta =D_{3}-D \end{array}$$
where *D*, *D*_2_ and *D*_3_ are the distance from the AE source position to sensors S_1_, S_2_ and S_3_, respectively. This method also must meet the velocities in the three directions that are the same in the lines of the AE source and the three sensors. In order to satisfy the above geometric relationship, the included angles of the propagation paths of the AE waves in the three directions must be satisfied.
(8)$$\{\begin{array}{c} \cos\Delta \theta_{12}\approx 1\\ \cos\Delta \theta_{13}\approx 1 \end{array}$$

When $\theta_{12}\leq \alpha ,\theta_{13}\leq \alpha$ are assumed, the error of $\cos\theta_{12},\cos\theta_{13}$ with 1 is acceptable, then
(9)$$\{\begin{array}{c} \frac{d\sin\theta }{D}\approx \sin(\Delta \theta_{12})\approx \tan(\Delta \theta_{12})\leq \tan\alpha \\ \frac{d\cos\theta }{d\sin\theta +D}\approx \sin(\Delta \theta_{13})\approx \tan(\Delta \theta_{13})\leq \tan\alpha \end{array}$$

The distance between the AE source and the sensor must satisfy the following formula:(10)$$D>\max(\frac{d\sin\theta }{\tan\alpha },\frac{d\cos\theta -d\tan\alpha \sin\theta }{\tan\alpha })$$

Using Equation (10), it can be found that the distance between the AE source and the sensor must be satisfied with greater than a certain value.

Then, the relationship between the error and the distance of the AE source and the sensors is studied. If the distance of the AE source and the sensor is *D*, the deviation of the position and the actual position is *R*, as shown in Figure 4. Additionally, if the deviation of the angle and actual angle is ∆*θ*, then
(11)$$R^{2}=D^{2}+D^{2}-2D^{2}\cos(\Delta \theta )$$
(12)$$R=D\sqrt{2[1-\cos(\Delta \theta )]}$$

From Equation (12), it can be found that, when the error caused by the angle is fixed, the error generated by the time-difference-blind localization method is proportional to the distance between the AE source and the sensor.

It can be determined that the reason for the error caused by the ideal waveform localization of the distribution of the wave speed in the composite plate according to the elliptical shape is mainly that the acoustic source position is not far enough from the sensor. It cannot satisfy that the wave velocities from the AE source to the three sensors are equal. At the same time, it cannot satisfy the condition $cos∆\theta_{12}\approx 1,cos∆\theta_{13}\approx 1$.

Using the horizontal comparison of the data in Table 2, it can be seen that the error is the smallest when the sensor spacing is 1 mm. This is because when the distance between sensors is 1 mm, the distance *d* between the sensors is very small compared to the distance *D* between the AE source and the sensor, which satisfies the hypothesis of that method. By the longitudinal comparison of the data in Table 2, it can be seen that the error increases with the increase of the distance between the AE source and the sensor. Therefore, by comparing the localization results of different sensor spacings, it can be verified that the reason for the errors in the above analysis of the time-difference-blind localization method is correct. However, since the sensor itself has a certain size, the minimum sensor spacing is the sum of the radii of the two sensors.

The time-difference-blind localization method experiences difficulty in meeting its assumptions in use, resulting in large errors in localization results. However, through analysis, it was found that the x-coordinate positioned by this method is accurate. Therefore, the approximate position of the acoustic emission source can be determined according to the abscissa of the localization result; for example, whether it is on the left or right of the composite material plate.

## 3. Principle and Performance Analysis of Beamforming Localization Method

The beamforming method can theoretically identify the location of any AE source in the near-field region. When the array is focused on a point source at limited distance, the incident AE waves are spherical, as shown in Figure 5. Array output is calculated by [15]
(13)$$b(\vec{r},t)=\frac{1}{M}\sum_{m=1}^{M}w_{m}x_{m}(t-\Delta_{m}(\vec{r}))$$
where $b(\vec{r},t)$ is the output of the array, and $\vec{r}$ is the direction vector from the reference sensor to the focused point. The reference point may be arbitrary, and it is the first sensor point on the left side in Figure 5; *M* is the number of sensors; $w_{m}$ the weighting coefficient for the channel of sensor m (usually no modification, in this article, $w_{m}\equiv 1$); $x_{m}(t)$ represents the signal acquired from the number *m* sensor; and $\Delta_{m}(\vec{r})$ indicates the individual time delay of the number *m* sensor to the reference point.

By adjusting time delay $\Delta_{m}(\vec{r})$, the signals associated with the spherical waves, emitting from the AE source focus, will be aligned in time before they are summed. As shown in Figure 5, $\Delta_{m}(\vec{r})$ can be obtained by
(14)$$\Delta_{m}(\vec{r})=\frac{\left|\vec{r}\right|-\left|\vec{r}-{\vec{r}}_{m}\right|}{c}$$where ${\vec{r}}_{m}$ is the distance between the reference point and number *m* sensor; $\left|\vec{r}\right|-\left|\vec{r}-{\vec{r}}_{m}\right|$ represents the difference between the distance from the reference point to the focus point and the distance from number *m* sensor to the focus point; and *c* is the propagation velocity of the AE wave.

The output of the beamforming method is the integral of $b(\vec{r},t)$ over time, which has the meaning of energy. If the focused point is the real source, the signals are aligned at the same wave front, and the output of the beamforming is maximum. However, the signals cannot be aligned at the same wave front when the array of sensors is focused on other locations, and the output of the beamforming is not the maximum. The location of the maximum output energy represents the location of the AE source. Under the beamforming method, every point on the structure is focused, and the output of the beamforming array at every point needs to be calculated. Once the location of the real AE source is focused on, the output of the sensor array reaches the maximum value. Thus, the location of the real AE source can be judged by comparing those outputs.

In operation, the localization accuracy can be determined according to actual needs. For example, on a 500 mm by 500 mm plate, if locating is performed with an accuracy of 10 mm, 2601 (51 × 51) points need to be scanned. If locating is performed with an accuracy of 1 mm, 251,001 (501 × 501) points need to be scanned. Obviously, the higher the localization accuracy, the more points need to be scanned. The beamforming localization method has high localization accuracy. However, the localization speed is very low since great quantities of points need to be scanned. The calculation speed is affected by the length of the calculation signal, the number of sensors, the localization accuracy (the higher the accuracy, the more points need to be scanned) and the size of the structure. Among these influencing factors, signal length, number of sensors, localization accuracy and structural size are all rigid requirements, and it is difficult to change them for the speed of calculation.

## 4. Joint Localization of the Time-Difference-Blind Localization Method and Beamforming Method

### 4.1. Principle of the Joint Localization Method

From Section 2 and Section 3, it can be concluded that the time-difference-blind localization method and the beamforming localization method have their own advantages and disadvantages. In this paper, these two methods are combined to develop a joint localization method, so that the localization error is smaller than that of the time-difference-blind localization method, and the localization speed is faster than that of the beamforming localization method.

The main idea of the joint localization method is: the time-difference-blind localization method takes less time to locate, and can determine the approximate location of the acoustic emission source, but the localization error is large, while the beamforming localization method takes a long time to locate, but the localization is accurate. Therefore, we firstly used the time-difference-blind localization method for preliminary localization, determining the approximate location of the acoustic emission source. Then, we used the beamforming localization method for precise localization.

The specific steps are:(1)Use the time-difference-blind localization method to locate the acoustic emission source with the collected data;(2)If the localization result of the time-difference-blind localization method is to the left (right) of the plate, bring the obtained data into the beamforming localization procedure. Change the area scanned in the program from the original whole plate to the left (right) half plate.

Therefore, the localization error generated by the joint localization method is the error generated by the beamforming localization method. This localization error is smaller than the error of the time-difference-blind localization method, and the localization time is reduced by half compared with the beamforming localization method.

### 4.2. Simulation Verification

The FE simulation model is an orthotropic plate with a size of 500 mm × 500 mm × 2 mm and the position and number of acoustic emission sources are consistent with those in Figure 2. The parameters are set as follows:$$E_{1}=21\;\mathrm{GPa},\;E_{2}=E_{3}=8.4\;\mathrm{GPa},\;G_{12}=G_{23}=4.78\;\mathrm{GPa},\;G_{13}=3.28\;\mathrm{GPa}\;\mu_{12}=\mu_{13}=0.3,\;\mu_{23}=0.28,\;\rho =1640\;\mathrm{Kg}/m^{3}$$

From the literature [18], the main parameters of the AE signal simulation are: the mechanical representation of the AE source, the time step, the size of the finite element and the sampling frequency. Because the AE signal of the simulation is the signal of a pencil lead break, it is independent of the material of the plate. Therefore, the simulated AE signal in the composite material is the same as the simulated AE signal in the steel plate, and the same mechanical expression is used. As shown in Figure 6, the chosen excitation time investigated is 2.5 μs. The sampling rate of 5 MHz is sufficient to resolve the observed signals frequency content in the range up to a maximal frequency of 0.2 MHz.

The size of the finite element is related to the minimum wavelength in the plate. If we want to obtain the complete wave signal, 20 nodes per wavelength are normally required. In the frequency range of interest, only the zero-order symmetric mode S_0_ and anti-symmetric mode A_0_ are present. These two modes are selectively excited in the model by applying appropriate nodal loads. For a maximum frequency of 200 kHz, the minimum wavelength is for A_0_ and it is given by
(15)$$\lambda_{\min}\approx \frac{c_{T}}{f_{\max}}$$

Considering a theoretical phase velocity of shear wave
(16)$$c_{{}_{T}}\approx \sqrt{\frac{G}{\rho }}$$
where *G* is the minimum value of the three shear modulus. $c_{T}=1414\;m/s$ and $\lambda_{\min}=7.07\;\mathrm{mm}$ can be obtained. In the present study, the value of $\lambda_{\min}$ was assigned as 10 mm, and S_0_ the element size was 0.5 mm. To avoid numerical instability, ABAQUS/EXPLICIT recommends a stability limit for the integration time step Δ*t* equal to:(17)$$\Delta t=\frac{1}{20f_{\max}}$$

To ensure that the information in the original signal is obtained, the sampling frequency must be greater than the maximum frequency of the signal. Additionally, the lowest criterion is that the sampling frequency is at least two times that of the maximum frequency of the signal.
(18)$$\Delta T=\frac{1}{2f_{\max}}$$

One can obtain Δt=0.25 μs and $\Delta T=2.5\times 10^{-6}\;s$. In this paper, the time step was assigned as 0.01 μs, and the sampling frequency was assigned as $2\times 10^{-7}\;s$. Simulation results are shown in Figure 7. The waveform of AE signal received by one sensor is shown in Figure 8.

As can be seen from Figure 7, the simulated stress cloud is elliptical. This proves that it is correct to distribute the velocity of the ideal waveform in each direction along the ellipse in the orthotropic plate. Because the main parameters of this method are the arrival times of waves at those sensors, the time of the first S_0_ wave peak of each sensor was defined as the arrival time in this paper. The time difference is the difference of the time corresponding to the first peak of the S_0_ wave received by each sensor.

The simulation data were located by using the time-difference-blind localization method, the beamforming localization method and the joint localization method. It should be noted that the accuracy produced by the step size of 10 mm in the beamforming localization method is within the acceptable range, so in this verification, a step size of 10 mm was used. In fact, the smaller the step size, the greater the effect of shortening the time. The localization results of the three methods are shown in Table 3. The localization errors of the three methods are shown in Table 4. The data in Table 3 and Table 4 are obtained by FE simulation.

From Table 3 and Table 4, it can be determined that the joint localization method can reduce the localization time, and the localization error is smaller than that of the time-difference-blind localization method. The localization error of the joint localization method is basically equal to the localization error of the beamforming localization method.

## 5. Experimental Verification

The experiment was conducted on a carbon fiber composite plate (long fiber braided composites) with a size of 500 mm × 500 mm × 2 mm, and the AE signal was generated by breaking a pencil lead [31]. The experimental setup is shown in Figure 9. It included AE sensors connected to the preamplifier to obtain the signal and the data acquisition instrument connected to a computer to analyze and process the signals. The sensor spacing was 50 mm, the coordinates of the six AE sensors were (100 mm, 50 mm), (50 mm, 100 mm), (100 mm, 100 mm), (400 mm, 50 mm), (400 mm, 100 mm) and (450 mm, 100 mm). Nine AE sources which coordinate are shown as the first column of Table 5. The pencil leads were broken at the same position three times. The experimental data were obtained by using the time-difference-blind localization method, the beamforming localization method and the joint localization method. A step size of 1 mm was used in the beamforming localization method. The localization results and errors of the three methods are shown in Table 5.

From Table 5, it can be concluded that the experimental results are consistent with the simulation results. The localization time of the joint localization method is shorter than that of the beamforming localization method, and the localization error is smaller than that of the time-difference-blind localization method. This proves that the proposed method improves the calculation speed of the beamforming localization method and reduces the localization error of the time-difference-blind localization method.

## 6. Conclusions

In this study, through the analysis of the localization performance of the time-difference-blind localization method and beamforming localization method, the advantages and disadvantages of the two methods were obtained. Based on this, a joint acoustic emission source localization method for composite materials was proposed. The superiority of the proposed method was verified by a simulation and experiment. Based on the investigation results, the following conclusions are drawn:(1)The assumption of the time-difference-blind localization method is that the position of the sound source is far enough from the sensors. However, the method shows difficulty in meeting its assumptions in use, resulting in large errors in localization results. Through analysis, it was found that the x-coordinate positioned by this method is accurate. Therefore, the approximate position of the acoustic emission source can be determined according to the abscissa of the localization result—for example, whether it is on the left or right of the composite material plate.(2)The beamforming localization method has high localization accuracy. However, the localization speed is very low since great quantities of points need to be scanned. The calculation speed is affected by the length of the calculation signal, the number of sensors, the localization accuracy (the higher the accuracy, the more points need to be scanned) and the size of the structure. Among these influencing factors, the signal length, number of sensors, localization accuracy and structural size are all rigid requirements, and it is difficult to change them for the speed of calculation.(3)Based on the advantages of the time-difference-blind localization method and beamforming localization method, the error of the joint localization method is smaller than the error of the time-difference-blind localization method, and the localization time is reduced by half compared with the beamforming localization method.

## Figures and Tables

**Figure 1 sensors-23-05473-f001:**
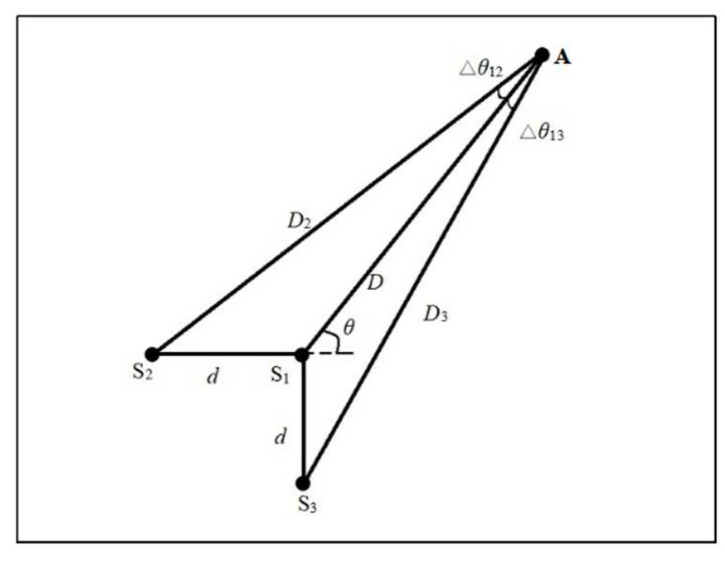
Geometric relationship of the time-difference-blind localization method.

**Figure 2 sensors-23-05473-f002:**
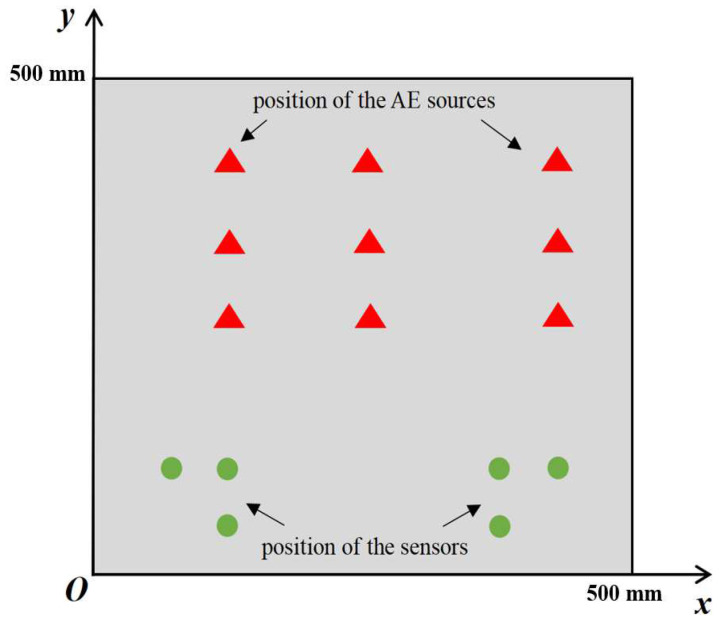
Position diagram of AE sources and sensors.

**Figure 3 sensors-23-05473-f003:**
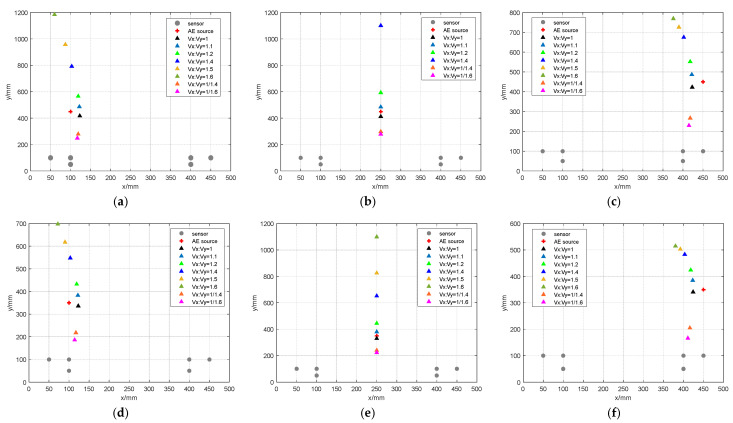
The localization results of different AE sources under different *Vx*:*Vy* values. (**a**) (100 mm, 450 mm); (**b**) (250 mm, 450 mm); (**c**) (450 mm, 450 mm); (**d**) (100 mm, 350 mm); (**e**) (250 mm, 350 mm); (**f**) (450 mm, 350 mm); (**g**) (100 mm, 250 mm); (**h**) (250 mm, 250 mm); (**i**) (450 mm, 250 mm).

**Figure 4 sensors-23-05473-f004:**
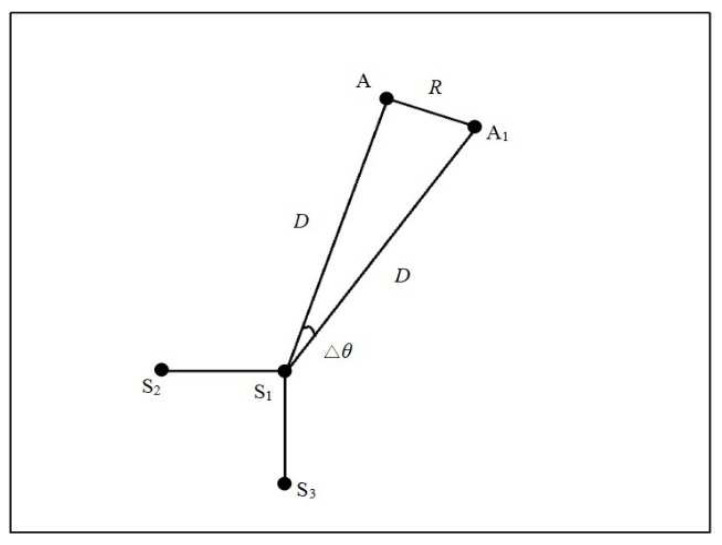
The relationship between error and distance.

**Figure 5 sensors-23-05473-f005:**
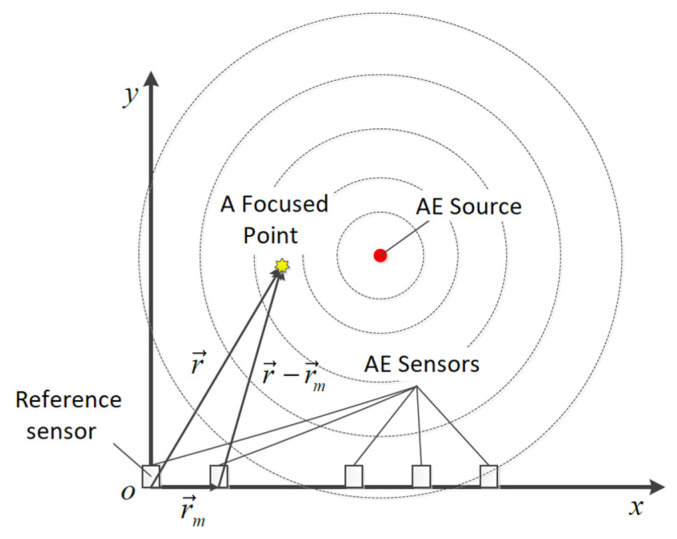
Schematic diagram of the principle of the beamforming method [17].

**Figure 6 sensors-23-05473-f006:**
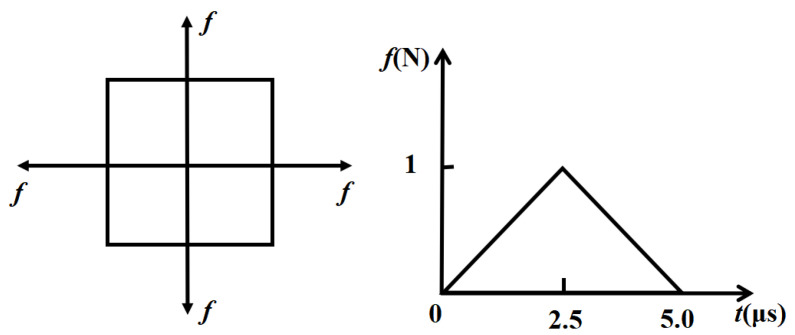
The acting forces and time function of the applied load.

**Figure 7 sensors-23-05473-f007:**
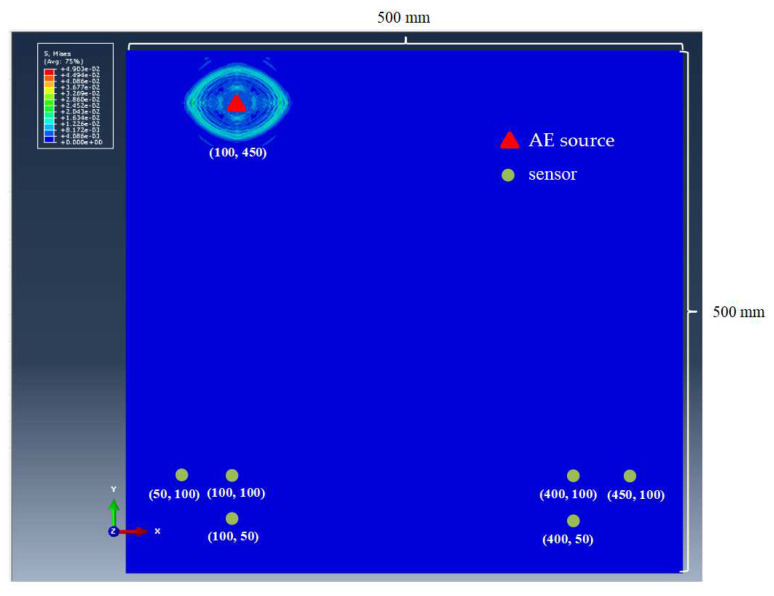
The FE simulation results of AE signal.

**Figure 8 sensors-23-05473-f008:**
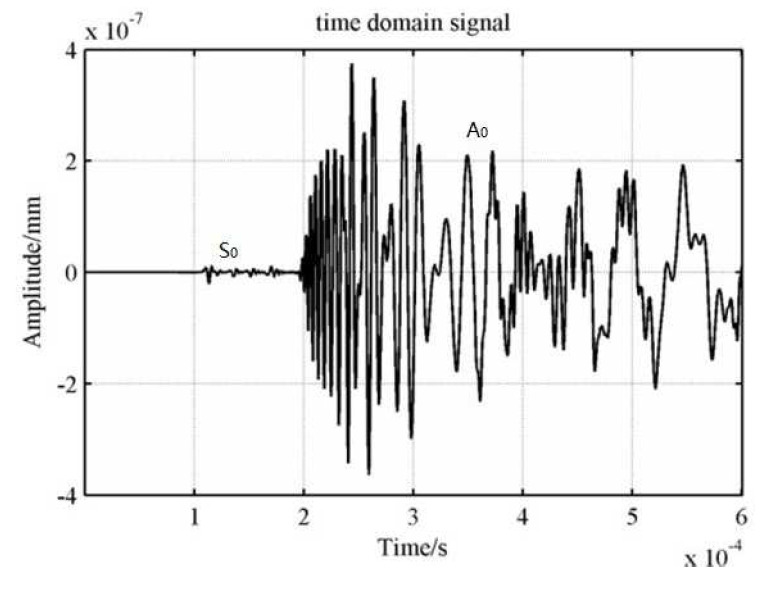
The waveform of AE signal received by one sensor.

**Figure 9 sensors-23-05473-f009:**
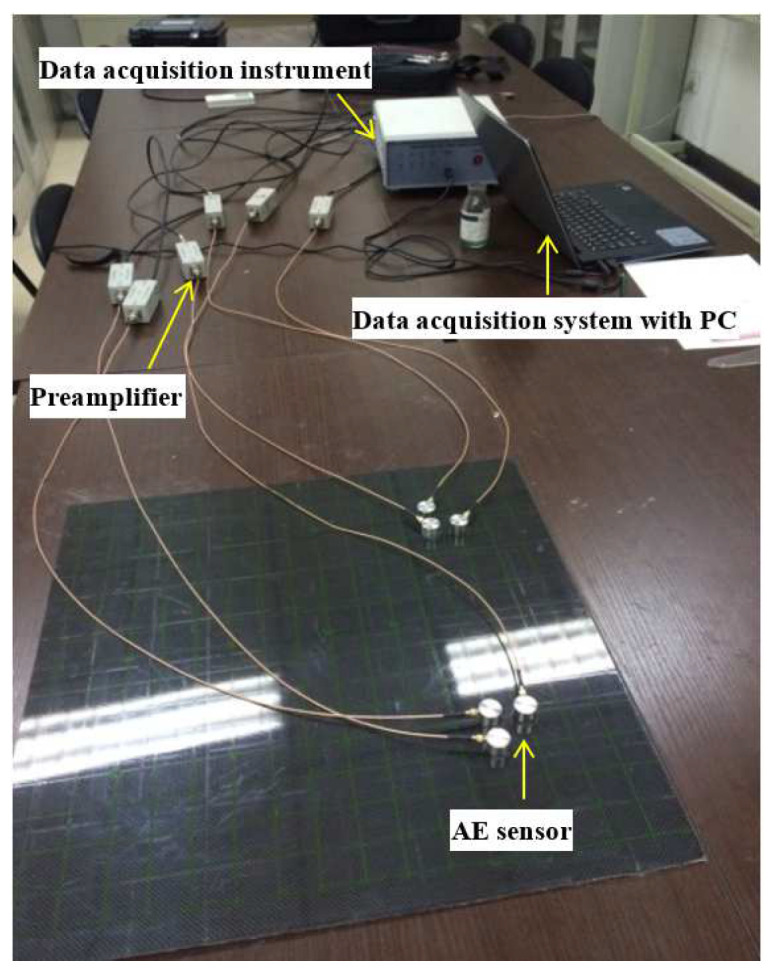
Experimental setup of the AE detection.

**Table 1 sensors-23-05473-t001:** Results of different *Vx*:*Vy* values.

**Coordinate** **of AE Source/mm**	***Vx*:*Vy* = 1**			***Vx*:*Vy* = 1.1**		
	**Locating Point/mm**	**Error of Coordinate/mm**	**Error of Distance/mm**	**Locating Point/mm**	**Error of Coordinate/mm**	**Error of Distance/mm**
(100, 250)	(125, 253)	(25, 3)	25	(123, 281)	(23, 31)	39
(250, 250)	(250, 250)	(0, 0)	0	(250, 279)	(0, 29)	29
(450, 250)	(427, 257)	(−23, 7)	24	(425, 284)	(−25, 34)	42
(100, 350)	(123, 336)	(23, −14)	27	(122, 383)	(22, 33)	40
(250, 350)	(250, 330)	(0, 30)	30	(250, 380)	(0, 30)	30
(450, 350)	(424, 341)	(−26, −9)	28	(423, 385)	(−27, 35)	44
(100, 450)	(123, 418)	(23, −32)	39	(122, 486)	(22, 36)	42
(250, 450)	(250, 412)	(0, −38)	38	(250, 485)	(0, 35)	35
(450, 450)	(423, 423)	(27, 27)	38	(422, 487)	(−28, 37)	46
**Coordinate** **of AE Source/mm**	***Vx*:*Vy* = 1.2**			***Vx*:*Vy* = 1.4**		
	**Locating Point/mm**	**Error of Coordinate/mm**	**Error of Distance/mm**	**Locating Point/mm**	**Error of Coordinate/mm**	**Error of Distance/mm**
(100, 250)	(120, 308)	(20, 58)	61	(105, 361)	(5, 111)	111
(250, 250)	(250, 309)	(0, 59)	59	(250, 376)	(0, 126)	126
(450, 250)	(420, 305)	(−30, 55)	63	(405, 331)	(−45, 81)	93
(100, 350)	(119, 433)	(19, 83)	85	(103, 548)	(3, 198)	198
(250, 350)	(250, 445)	(0, 95)	95	(250, 652)	(0, 302)	302
(450, 350)	(418, 424)	(−32, 74)	81	(403, 483)	(−47, 133)	141
(100, 450)	(119, 566)	(19, 116)	118	(103, 792)	(3, 342)	342
(250, 450)	(250, 593)	(0, 143)	143	(250, 1101)	(0, 651)	651
(450, 450)	(418, 552)	(−32, 102)	107	(402, 675)	(−48, 225)	230
**Coordinate** **of AE Source/mm**	***Vx*:*Vy* = 1.5**			***Vx*:*Vy* = 1.6**		
	**Locating Point/mm**	**Error of Coordinate/mm**	**Error of Distance/mm**	**Locating Point/mm**	**Error of Coordinate/mm**	**Error of Distance/mm**
(100, 250)	(97, 388)	(−3, 138)	138	(81, 417)	(−19, 167)	168
(250, 250)	(250, 410)	(0, 160)	160	(250, 445)	(0, 195)	195
(450, 250)	(396, 337)	(−54, 87)	102	(386, 341)	(−64, 91)	111
(100, 350)	(90, 618)	(−10, 268)	268	(72, 698)	(−28, 348)	349
(250, 350)	(250, 826)	(0, 476)	476	(250, 1099)	(0, 749)	749
(450, 350)	(392, 503)	(−58, 153)	164	(380, 515)	(−70, 165)	179
(100, 450)	(87, 957)	(−13, 507)	507	(60, 1185)	(−40, 635)	636
(250, 450)	(250, 1933)	(0, 1483)	1483	(250, 7663)	(0, 7213)	7213
(450, 450)	(390, 726)	(−60, 276)	282	(376, 769)	(−74, 319)	327
**Coordinate** **of AE Source/mm**	***Vx*:*Vy* = 1/1.4**			***Vx*:*Vy* = 1/1.6**		
	**Locating Point/mm**	**Error of Coordinate/mm**	**Error of Distance/mm**	**Locating Point/mm**	**Error of Coordinate/mm**	**Error of Distance/mm**
(100, 250)	(113, 157)	(13, −93)	94	(106, 121)	(6, −129)	129
(250, 250)	(250, 181)	(0, −69)	69	(250, 165)	(0, −85)	85
(450, 250)	(412, 127)	(−38, −123)	129	(402, 106)	(−48, −144)	152
(100, 350)	(117, 218)	(17, −132)	133	(114, 186)	(14, −164)	165
(250, 350)	(250, 240)	(0, −110)	110	(250, 223)	(0, −127)	127
(450, 350)	(416, 205)	(−34, −145)	149	(411, 166)	(−39, 184)	188
(100, 450)	(119, 280)	(19, −170)	171	(117, 249)	(17, −201)	202
(250, 450)	(250, 298)	(0, 152)	152	(250, 278)	(0, −172)	172
(450, 450)	(418, 267)	(−32, −183)	186	(415, 230)	(−35, −220)	223

**Table 2 sensors-23-05473-t002:** Different sensor spacing localization results.

Coordinate of AE Source/mm	Error/mm			
	Sensors Spacing			
	1 mm	10 mm	30 mm	50 mm
(100, 450)	0.86	8.48	24.35	38.89
(250, 450)	0.90	8.75	24.47	38.15
(450,450)	0.81	8.00	23.39	37.94
(100, 350)	0.58	5.74	16.83	27.34
(250, 350)	0.48	4.62	12.88	20.02
(450, 350)	0.54	5.43	16.43	27.43
(100, 250)	0.51	5.1	15.19	24.88
(250, 250)	0	0	0	0
(450, 250)	0.45	4.59	14.27	24.51

**Table 3 sensors-23-05473-t003:** The localization results of the three methods.

Coordinate of AE Source/mm	Time-Difference-Blind Localization Method		Beamforming Localization Method		Joint Localization Method	
	Locating Point/mm	Calculating Time/s	Locating Point/mm	Calculating Time/s	Locating Point/mm	Calculating Time/s
(100, 450)	(120, 833)	0.373936	(100, 480)	3.789047	(100, 480)	2.268460
(250, 450)	(250, 787)	0.796325	(250, 150)	3.897832	(250, 150)	2.745241
(450, 450)	(421, 846)	0.110530	(450, 480)	3.784312	(450, 480)	2.002686
(100, 350)	(125, 649)	0.107162	(100, 370)	3.712398	(100, 370)	1.963361
(250, 350)	(250, 609)	0.813859	(250, 340)	3.732149	(250, 340)	2.679934
(450, 350)	(427, 692)	0.118872	(450, 360)	3.873213	(450, 360)	2.055478
(100, 250)	(130, 468)	0.106911	(120, 390)	3.897124	(120, 390)	2.055473
(250, 250)	(250, 442)	0.840809	(250, 250)	3.984321	(250, 250)	2.832970
(450, 250)	(434, 497)	0.099531	(470, 430)	3.789047	(470, 430)	2.021392

**Table 4 sensors-23-05473-t004:** The localization errors of the three methods.

Coordinate of AE Source/mm	Time-Difference-Blind Localization Method		Beamforming Localization Method		Joint Localization Method	
	Error of Coordinate/mm	Error of Distance/mm	Error of Coordinate/mm	Error of Distance/mm	Error of Distance/mm	Error of Distance/mm
(100, 450)	(20, 383)	384	(0, 30)	30	(0, 30)	30
(250, 450)	(0, 337)	337	(0, 300)	300	(0, 300)	300
(450, 450)	(29, 396)	397	(0, 30)	30	(0, 30)	30
(100, 350)	(25, 299)	300	(0, 20)	20	(0, 20)	20
(250, 350)	(0, 259)	259	(0, 10)	10	(0, 10)	10
(450, 350)	(23, 342)	343	(0, 10)	10	(0, 10)	10
(100, 250)	(30, 218)	220	(20, 140)	141	(20, 140)	141
(250, 250)	(0, 192)	192	(0, 0)	0	(0, 0)	0
(450, 250)	(16, 247)	248	(20, 180)	181	(20, 180)	181

**Table 5 sensors-23-05473-t005:** The localization results and errors of the three methods.

Coordinate of AE Source /mm	Time-Difference-Blind Localization Method			Beamforming Localization Method			Joint Localization Method		
	Locating Point/mm	Error of Coordinate/mm	Error of Distance/mm	Locating Point/mm	Error of Coordinate/mm	Error of Distance/mm	Locating Point/mm	Error of Coordinate/mm	Error of Distance/mm
(100, 450)	(140, 298)	(40, 152)	157	(102, 477)	(2, 27)	27	(102, 477)	(2, 27)	27
(250, 450)	(237, 301)	(13, 149)	150	(253, 407)	(3, 43)	43	(253, 407)	(3, 43)	43
(450, 450)	(462, 418)	(12, 32)	34	(444, 466)	(6, 16)	17	(444, 466)	(6, 16)	17
(100, 350)	(138, 266)	(38, 84)	92	(100, 366)	(0, 16)	16	(100, 366)	(0, 16)	16
(250, 350)	(235, 256)	(15, 94)	95	(252, 347)	(2, 3)	4	(252, 347)	(2, 3)	4
(450, 350)	(463, 373)	(13, 23)	26	(449, 361)	(1, 11)	11	(449, 361)	(1, 11)	11
(100, 250)	(147, 238)	(47, 12)	49	(118, 395)	(18, 145)	146	(118, 395)	(18, 145)	146
(250, 250)	(234, 220)	(16, 30)	34	(250, 249)	(0, 1)	1	(250, 249)	(0, 1)	1
(450, 250)	(466, 302)	(16, 52)	54	(475, 442)	(25, 192)	194	(475, 442)	(25, 192)	194

## Data Availability

The data can be got in this paper.

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
