# Peer review of "A Joint Acoustic Emission Source Localization Method for Composite Materials"

_sensors, 2023, doi:10.3390/s23125473_

Round 1
Reviewer 1 Report (Previous Reviewer 1)
The authors have made significant changes to the manuscript. a few minor changes are suggested below:
1. Table 3 and 4 show error of coordinates, which is not a quantitative comparison metric. Instead i recommend converting it into error% for easier comparison.
2. Table 3 and 4 should clearly state that it is uses the FE simulation.
3. Fig 7: The authors are showing the wave propagation snapshot, but fail to show the receiver locations, distances etc. in the plot. Apart from showing that the authors can use FE software, the figure is not conveying any other information.
4. There are some sentence formation errors which can be eliminated by a close edit.
There are some sentence formation errors which can be eliminated by a close edit.
Author Response
Please see the attachment.

Reviewer 2 Report (New Reviewer)
The Authors propose a joint application of the time difference blind localization method and beamforming localization method aimed at the optimization (both in terms of accuracy and calculation amount) of AE source location in composite plate structures. The research work is interesting and well-balanced between simulation and experiment. Its publication is recommended by addressing the following concern:
(1) Please add dimensions in the x and y directions on Figure 2.
(2) In Eq(13), in which way the expression xm(t − Δm(r)) is obtained?
(3) In line 237, the weighting coefficients wm is equal to 1, why?
(4) What are the meanings of the abbreviations ‘S0’ and ‘A0’ in Figure 8?
Minor editing of English language required.
Author Response
Please see the attachment.

Reviewer 3 Report (New Reviewer)
By analyzing the performance of the time difference blind localization method and beamforming localization method, the manuscript proposes a joint acoustic emission source localization method for composite materials. And the superiority of the proposed method is verified through simulation and experiments. Overall, the paper is interesting and well written. However, the following issues and remarks should be taken into account when revising the paper before re-consideration for publication is made.
(1) In the introduction, from the perspective of significant progress in the research and use of composite materials in various fields of industry in recent years, I believe that past literature is outdated. Therefore, I request an increase in research work, which will further clarify the current research trends in composite material testing.
(2) In the introduction, the authors stated 'Wang et al [11] proposed a joint localization method based on beamforming and time difference of arrival '. What are the differences between the present paper and Wang et al. [11]? They should be clearly stated in the paper, otherwise it is difficult to appreciate the contributions of the present study.
(3) in Line 355-356: The author provides that the plate used in the experiment is a carbon fiber composite plate. How many layers does this plate have? What is the fiber direction?
Author Response
Please see the attachment.

This manuscript is a resubmission of an earlier submission. The following is a list of the peer review reports and author responses from that submission.
Round 1
Reviewer 1 Report
This article presents an analysis of the time difference blind localization method proposed earlier. While the objective is of interest to the community, the article has some major issues, which does not address the key question.
1. The authors assume the anisotropy purely works based on Vx/Vy ratio, which is not correct. This is especially true for composite plates where for a unidirectional laminate, Vx not equal to Vy, but for a cross ply laminate, vx = vy, but not equal to v@45 degree. So reducing anisotropy to a simple case of vx = vy is not sufficient.
2. The simulation results are somewhat confusing: I am confused on what fig 4 is supposed to be showing. Same goes for fig. 5 where the FFT of a broad time domain signal is shown, which includes both S0 and A0 modes. I am not sure what the focus is.
3. The authors identify three main sources of error: distance between sensors, anisotropy, and sensor spacing. However, the analysis and data presentation seems to have little relevance to these parameters. The motivation for the study is not fully clear. The authors want to do an in-depth analysis, but their experiments design is not coherent with the objective. They only chose 1 composite plate, where the anisotropy is constant. I recommend redesigning it such that the parameters of interest are explored and conclusions are drawn coherent with the observations.
4. The data organization and presentation can be improved. For example: the table data can be converted into location images as shown in Ref. 20.
5. if a carbon fiber plate was used, then the properties in line 219 will not match with experimental conditions. While the objective is a blind localization, the lack of coherency between experiments, model and method used cannot be overlooked.
6. The conclusions are very qualitative and i am not sure why we need these, when its very apparent in Ref. 20. I recommend reorganizing the data to draw quantitative conclusions.
Author Response
Please see the attachment。

Reviewer 2 Report
This paper deals with a very important but, also, a delicate subject to deliver accurate information. Acoustic Emission technique has been widely used in materials characterization focusing greatly in composite materials since, due to its heterogeneity, the fracture processes in these materials are extremely complex. Talking about “not Knowing Material Properties” is somehow risky since it is very well known that, for each design of composite materials, matrix, geometry and type of reinforcement, adhesion of the fiber/matrix interface, volume fraction, etc., the mechanical response to the load exerted on the composite material will vary according to these factors. Even current analytical models are not capable of encompassing the full range of mechanisms that govern the fracture process. Therefore, I consider that new methodologies for assessing failure mechanisms are vital to understand in order to identify such mechanisms from damage initiation until total fracture and the factors that provoke them.
There are some doubts I would like the authors to clarify:
1. There are some English writing mistakes in the manuscript. I understand, as a no native English speaker myself, it is difficult to write as clear as possible in a foreign language, so I recommend the paper to be checked by an English speaker. For example, try use synonyms as much as possible avoiding repeat the same word many times. Check punctuation, etc. I have pointed out some errors in main manuscript attached (PDF).
2. For the first analysis an ideal composite plate is assumed and gives some information regarding AE waves behavior on it. However, it is important to define “ideal composite plate”, it is a homogenous material? Isotropic or anisotropic? Metal, ceramic, polymeric? Reinforced? Some readers may not understand. Therefore, my suggestion is to clarify what type of ideal material are the authors assuming and define it.
3. Authors state (page 05, lines 152) that: “Localization error increases with the increase of the distance between the acoustic source and the sensor.” Regarding AE waves traveling in a material some phenome are needed to be taken in account such as attenuation, diffraction and scattering phenomenon. Were these parameters considered in your analysis?
4. Again please define what authors meant by “an inhomogeneous plate” (page 08, lines 218).
5. Authors use a “carbon fiber composite plate” for experimental analysis. Please give details of this composite; for example: long unidirectional carbon fibers? Short fibers? textile? Fiber volume fraction? All these factors can change the results for AE wave velocity.

Reviewer 3 Report
The work is devoted to the topical topic of localization of acoustic emission sources. The results are of practical interest and can be published. As a recommendation, I would like to get data on the relationship of the accuracy of determining coordinates with the amplitude and frequency of the acoustic emission signal
Round 2
Reviewer 1 Report
The author have revised the article. However, the their response does not address the central issues i had raised.
1. The authors claim they only approximate the anisotropy. However, that was already carried out by Kundu, so i am not sure the significance of the present work. For orthotropic materials like the CFRP, that approximation is not acceptable.
2. The issue with Fig. 4, i cannot see anything in the figure apart from a blue screen. The figure quality is very bad and i recommend changing the figure to depict the features of interest.
3. I agree that the article explores several analytical cases with some approximations. However, there is no validation for them except for 1 case. I recommend changing the structure of the article to show a more systematic approach and clearly discuss the effect of each parameter.
4. The authors claim the composite plate has unknown properties, which does not make sense. The experimental design should have first measured the properties and carried out a blind localization. However, this was never validated, which is one of the issues with the present study.
5. Once again, the conclusions seem to be recommendations rather than quantitative observations. Given that the authors wanted to carry out comprehensive parametric analysis, the conclusions dont seem to support it. I recommend developing quantitative relationships between the different parameters to make this study reach wider audience. Presently, the study is very qualitative and seems to simply reinforce Kundu et al.'s work.
